# A Self-Deformation Robot Design Incorporating Bending-Type Pneumatic Artificial Muscles

**Hiroki Tomori [1],\* , Kenta Hiyoshi [1], Shonosuke Kimura [1], Naoya Ishiguri [2] and Taisei Iwata [2]**

[1]  Department of Mechanical Systems Engineering, Graduate School of Science and Engineering, Yamagata University, Yamagata 992-8510, Japan
[2]  Department of Mechanical Systems Engineering, Faculty of Engineering, Yamagata University, Yamagata 992-8510, Japan
\*  Correspondence: tomori@yz.yamagata-u.ac.jp; Tel.: +81-238-26-3217

**Abstract:** With robots becoming closer to humans in recent years, human-friendly robots made of soft materials provide a new line of research interests. We designed and developed a soft robot that can move via self-deformation toward the practical application of monitoring children and the elderly on a daily basis. The robot's structure was built out of flexible frames, which are bending-type pneumatic artificial muscles (BPAMs). We first provide a description and discussion on the nature of BPAM, followed by static characteristics experiment. Although the BPAM theoretical model shares a similar tendency with the experimental results, the actual BPAMs moved along the depth direction. We then proposed and demonstrated an effective locomotion method for the robot and calculated its locomotion speed by measuring its drive time and movement distance. Our results confirmed the reasonability of the robot's speed for monitoring children and the elderly. Nevertheless, during the demonstration, some BPAMs were bent sharply by other activated BPAMs as the robot was driving, leaving a little damage on these BPAMs. This will be addressed in our future work.

**Keywords:** pneumatic; artificial muscle; soft robot

## 1. Introduction

Conventional robots are typically industrial robots with a separate workspace from human workers. This way, the robots could increase their own stiffness, improve their work accuracy, and become faster in completing their tasks. By contrast, robots developed recently have become closer to humans. Because such types are often in contact with humans, they should be designed to secure high safety for humans and the surroundings. Nevertheless, unlike industrial robots, human-friendly robots assume a complicated life space. For instance, they are installed with sensors for depth, force, and torque to help them avoid collisions with obstacles or reduce the damage dealt [1,2]. However, technologies for robot recognition of a complicated environment have high calculation costs and may not prevent all unintended collisions. Therefore, these robots need constructive safety, i.e., they need to protect themselves in case of collision with humans or the environment.

Soft robotics is currently an attractive research interest highlighting this need. Soft robots tend to be lightweight and flexible, which is ideal for constructive safety [3]. Suzumori developed an arm consisting of an air balloon to achieve height safety [4]. Ansari developed a manipulator using McKibben actuators for the personal care of elder people [5]. Ward-Cherrier developed a soft tactile sensor that allows a robot to interact with humans safely [6]. Furthermore, these characteristics of a soft robot also improves performance. Ortiz [7] suggested that a robot's bioinspired motion improves locomotion performance on ground-like granular substrates. Especially, to achieve the bioinspired motion, they developed soft robot digging through dry granular ground. Furthermore, Suzumori [8]

developed a pipe inspection robot using flexible microactuators, which can adapt to a narrow and complex space without complicated control. Hawkes developed a soft robot that can navigate in a narrow space by growing [9]. Such a behavior is possible because these robots have a soft construction. In addition, soft skins that contribute to robots increasing environment adaptability [10] and grasping ability [11] were developed. In this manner, soft robots have advantages in terms of not only safety for human and the surrounding environment, but also in terms of performance such as the adaptability toward complicated and indeterminate environments, and in reducing calculation costs in control and the grasping ability of the manipulator. As such, research about soft sensors [12] will accelerate the development of soft robots. We also consider that flexible touch and motion reduce the chance of an intimidating impression to the user. Thus, we aim to develop a soft-material, human-friendly robot incorporating these features and expect it to be applied for the daily monitoring of people, especially children and the elderly. This robot needs to have high safety and adaptability to a human's life environment.

Configuration-wise, these robots are normally driven by soft actuators made of soft materials such as rubber, gel, or a polymer [13,14]. Though flexible actuators tend to be less powerful, accurate, and responsive than high-stiffness actuators, their flexibility offers advantages in terms of environmental adaptability. Commonly used soft actuators include ionic polymer-metal composites (IPMC) [15,16], dielectric elastomer actuators (DEA) [16], conducting polymer [17], shape memory actuator (SMA) [18], etc. Although they have characteristics as a soft actuator, there are various advantage and disadvantages. Kongahage [19] had compared characteristics between soft actuators. In general, the strain tolerance of these actuators is low as a whole, yet a soft robot needs a practical output and strain in order to work. Among them, SMA, DE, and a pneumatic actuator can possibly to satisfy the requirements. Additionally, considering that SMA and DE need heating and high voltage, respectively, we concluded that a pneumatic actuator is suitable for a daily monitoring robot. A pneumatic actuator consists of fibers and a flexible membrane like rubber. It actuates via deformation (expansion and contraction), similar to muscles, upon the application of air pressure. The actuator has no sliding or moving parts, such as a pneumatic cylinder [20], which means some advantages such as dust proofing, high sealability, and low noise. Also, its working fluid (air) is environment- and user-friendly. Furthermore, the pneumatic actuator's flexibility and lightness can be used for a highly human-friendly actuator while it generates a high output relative to other soft actuators. Thus, its high safety encourages the application of robots toward working nearby humans, for example, in the rehabilitation field [21,22].

Besides using pneumatic rubber actuators, we designed a simple structure and shape of the soft robot to make it sturdy but lightweight, and to especially focus on its flexibility. This is a different scheme from previous pneumatic-rubber-actuator-based typical robots that incorporate a rigid mechanism to transfer the actuator's output into a link. For example, the wire-and-pulley mechanism can translate the contraction of a pneumatic rubber artificial muscle to the pulley rotation [23,24], but can show flexibility only in the rotation direction of the pulley. We addressed this problem with the proposed self-deformation robot (SDR) with an integrated soft frame mechanism. The robot is constructed from pneumatic rubber actuators as soft frames and moves by deforming itself. We also developed a bending-type pneumatic rubber artificial muscle (BPAM) for frames. A BPAM is a rubber tube constrained by fibers that bends with the application of air pressure. Its basic principle is the same as that described by Miyagawa [25]; in this reference, Miyagawa presented a tube-shape pneumatic soft actuator that creates a bending motion. Then, they also investigated static characteristics of that actuator using the principle of virtual work. Although the cross-section shape of their actuator differs from our BPAM, the procedure for the derivation of an equation model was applicable. This actuator does not impair the flexibility of the SDR because it can produce a bending motion without any rigid mechanism.

We first explain the principle of BPAM and discuss the proposed SDR design. Afterward, we experimentally demonstrate the SDR locomotion. We present the general findings of this study in the last section.

## 2. Bending-Type Pneumatic Rubber Artificial Muscle

### 2.1. Mechanism of the BPAM

Photographs of the BPAM are displayed in Figure 1. This actuator consists of a rubber tube and an aramid fiber, and bends by applying air pressure. Here, the drive principle of BPAM is explained with Figure 2. In Figure 2, a rubber tube (A) expands in all directions (B) with air pressure by Pascal's principle. Therefore, the aramid fiber rolled in the rubber tube without space restraints expands in a radial direction (C). In this case, the tube extends in only the axial direction by applying air pressure (D). In addition, an aramid fiber of 8 mm width was utilized to limit the axial length of a part of the tube walls (E). Due to this mechanism, the difference of extension in the tube upon the application of air pressure produces bending of the BPAM in the direction where the axial length is constrained (F).

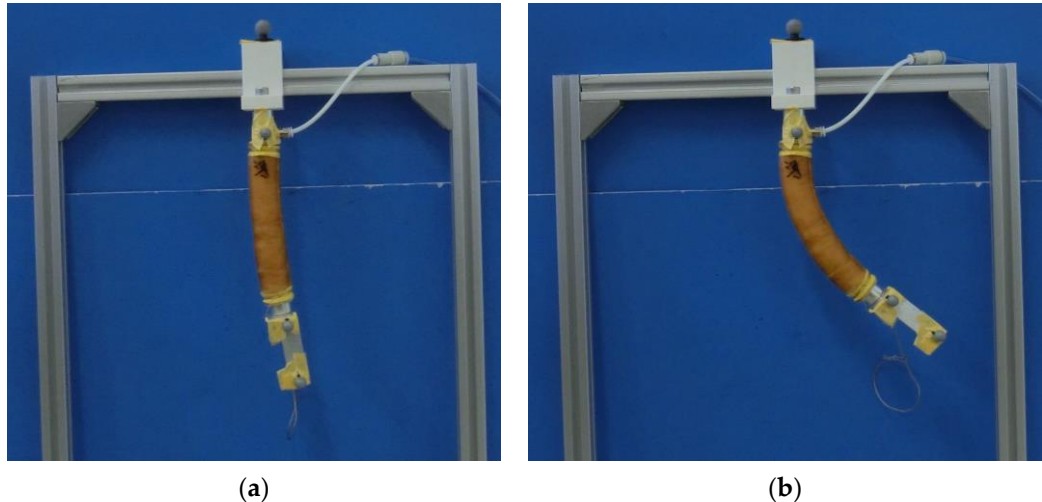

|           |           |
|-----------|-----------|
| (**a**)   | (**b**)   |

**Figure 1.** A bending-type pneumatic artificial muscle: (**a**) air is released, and (**b**) air is applied.

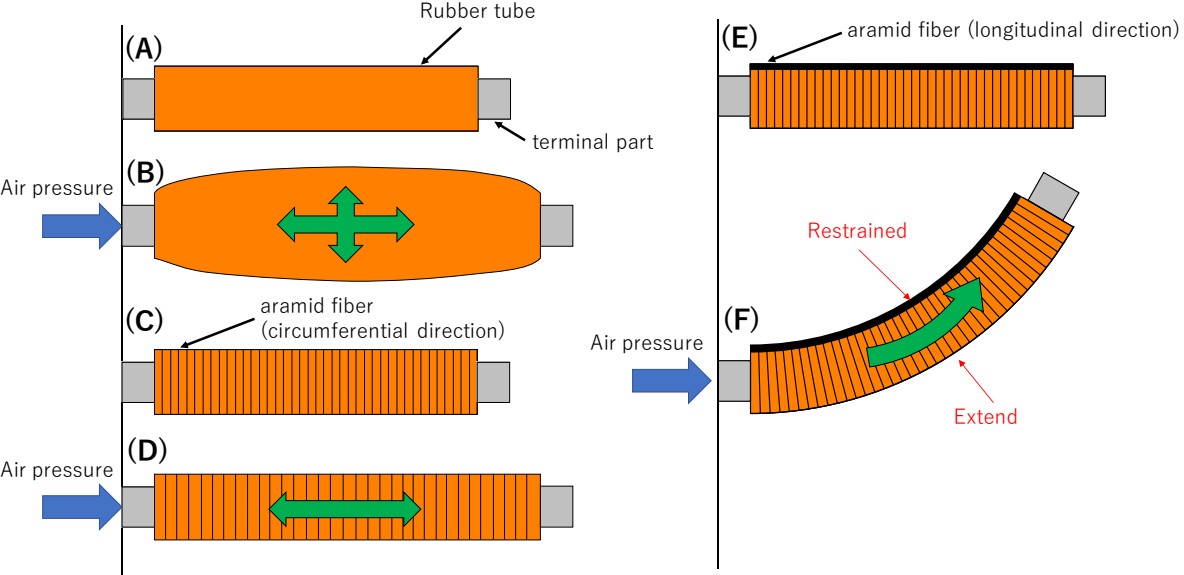

**Figure 2.** Mechanism of bending-type pneumatic artificial muscles (BPAM).

Then, a schematic of the BPAM is shown in Figure 3 and the specifications of the BPAM are listed in Table 1. In Figure 3, we assumed the BPAM bends in the plane and load *F* was applied to BPAM's end orthogonally, as indicated by the red vector.

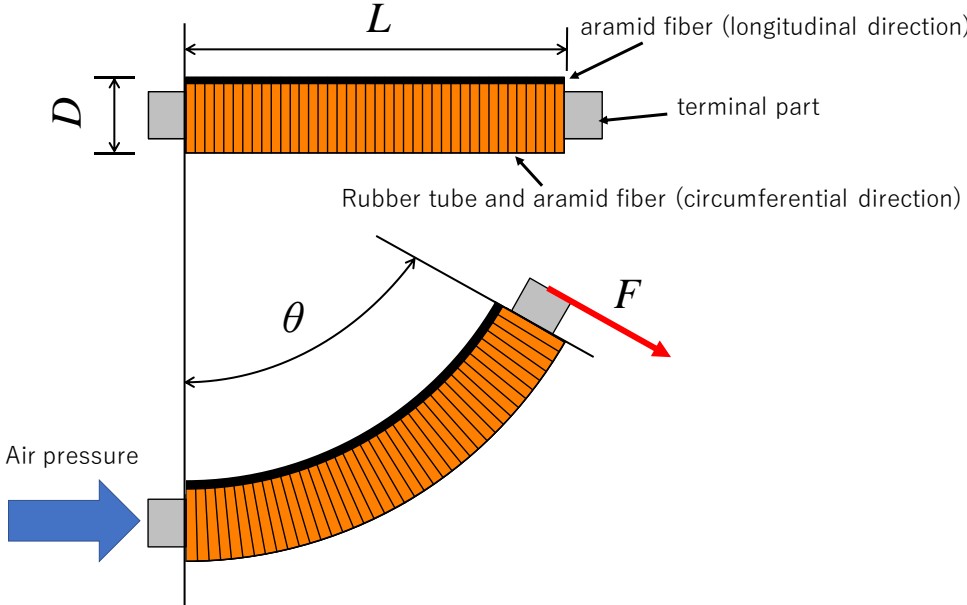

**Figure 3.** Definition of dimension of BPAM.

**Table 1.** Bending-type pneumatic artificial muscles (BPAM) specifications.

| Parameter | Specification |
|---|---|
| Length (mm) | 100 |
| Outer diameter (mm) | 21 |
| Inner diameter (mm) | 16 |
| Weight (kg) | 0.05 |

## 2.2. Static Characteristics Model of BPAM

Here, we provide an explanation of the static characteristics of the BPAM. In this section, we obtained equation models based on the principle of virtual work with reference to Miyagawa's work [25]. An equilibrium between applied air pressure $P$ (Pa), load $F$ (N), and bending angle $\theta°$ is illustrated via equations. It is used not only for understanding BPAM's characteristics, but also for force control and simulation as a future step.

Figure 4 shows a view of BPAM's end part. Furthermore, Figure 5 shows a schematic view of a bending BPAM. In Figure 4, the force $F(r, \phi)$ acting on a small area $dS$ at position $(r, \phi)$ of the surface of the terminal part by air pressure, along with a moment around the $x$-axis $M_f$, are provided in Equations (1) and (2):

$$F(r, \phi) = PdS \tag{1}$$

$$\begin{aligned} M_f &= \int_S F(r,\phi)(r\,sin\phi + t)dS \\ &= P\pi\left(a^3 + a^2 t\right) \end{aligned} \tag{2}$$

where $\alpha$ is the inner radius of the tube (mm) and $t$ is the thickness of the tube (mm).

Then, we obtained a moment around the $x$-axis using the elastic restoring force of rubber. The elastic restoring force $q_r(\phi)$ acting at position $\phi$ on a cross-section surface of rubber tube is shown in Equation (3):

$$q_r(\phi) = -\frac{Eat}{L}\theta(r_c\,sin\phi + t)d\phi \tag{3}$$

$$r_c = a + \frac{t}{2} \tag{4}$$

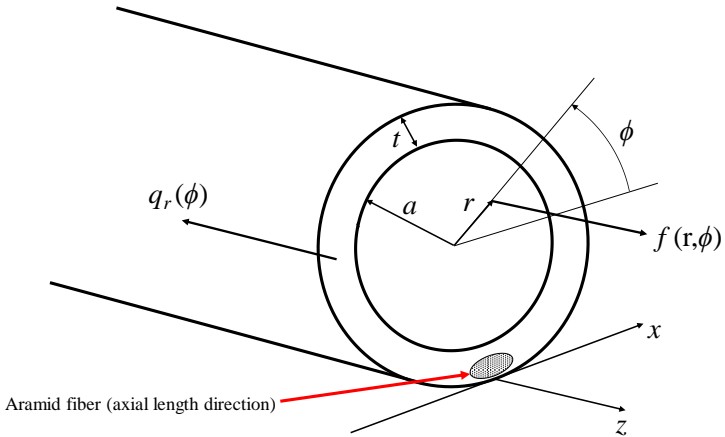

**Figure 4.** Schematic view of BPAM's end part.

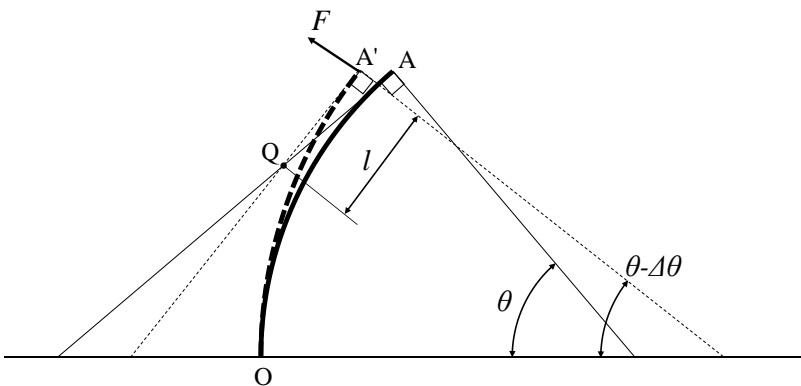

**Figure 5.** Schematic view of a bending BPAM.

where $L$ is the length of the tube (mm), $r_c$ is the representative radius of the tube (mm), and $E$ is Young's modulus in the axial direction of the rubber. Here, we assumed that the cross-section shape of BPAM does not change during bending and the thickness $t$ of the tube is constant. Thus, we considered Young's modulus $E$ at position $(r, \phi)$ varies according to Equation (5):

$$E = \frac{E_0 L}{L + \theta(r_c \sin\phi + a + t)} \tag{5}$$

where $E_0$ is the Young's modulus of rubber before deformation (Pa). From Equations (3)–(5), the moment around the $x$-axis $M_{qr}$ due to the elastic restoring force is obtained as Equation (6):

$$M_{qr} = \int_0^{2\pi} q_r(\phi)\ (r_c \sin\phi + a + t)$$
$$= -E_0 a t \theta \int_0^{2\pi} \frac{(r_c \sin\phi + a + t)^2}{L + (r_c \sin\phi + a + t)\theta} d\phi \tag{6}$$

Next, we considered the equilibrium condition between the external force, air pressure, and elastic restoring force. The external force acting on the tip of the BPAM, as seen in Figure 5, equilibrates with the force generated by changing the bending angle $\theta$. We considered a situation where the bending angle $\theta$ changed from this equilibrium condition. In this case, the total work by the applied air pressure, elastic restoring force, and external force must be zero because of the principle of virtual work. In Figure 5, the external force $F$ creates a moment around point Q. Furthermore, the distance between point Q and the point of application of force A' is given as Equation (7):

$$l = \frac{L}{\theta^2}(1 - \cos\theta) \qquad (\Delta\theta \ll \theta) \tag{7}$$

Additionally, work $W_f$ by the moment from external force $F$ is obtained as Equation (8):

$$W_f = -Fl\Delta\theta = -F\frac{L}{\theta^2}(1 - cos\theta)\Delta\theta \tag{8}$$

On the other hand, work $W_c$ done by moment $M_f$ and $M_{qr}$ is shown in Equation (9):

$$W_c = \left(M_f + M_{qr}\right)\Delta\theta \tag{9}$$

Finally, the total work must be zero, as in Equation (10), because of the principle of virtual work. From this equation, the relationship between applied air pressure $P$, load $F$, and bending angle $\theta$ is illustrated as Equation (11):

$$-F\frac{L}{\theta^2}(1 - cos\theta)\Delta\theta + \left(M_f + M_{qr}\right)\Delta\theta = 0$$
$$F = \frac{M_f + M_{qr}}{\frac{L}{\theta^2}(1 - cos\theta)} \tag{10}$$

$$P = \frac{F\frac{L}{\theta^2}(1 - cos\theta) - M_{qr}}{a^3\pi + a^2t\pi} \tag{11}$$

### 2.3. Static Characteristics Experiments

We conducted a static characteristics experiment to determine the fundamental characteristics of the BPAM. The schematic for obtaining the relationships between air pressure, load, and bending angle is depicted in Figure 6. Bending angle was measured using image analysis with motion capture systems. In this experiment, air pressure was applied to the BPAM starting from a 0.02 baseline and increased in increments of 0.02 MPa, whereas load $F_1$ was increased in increments of 0.1 kg for a range of 0–0.5 kg. Load $F$ was calculated as the horizontal component of $F_1$.

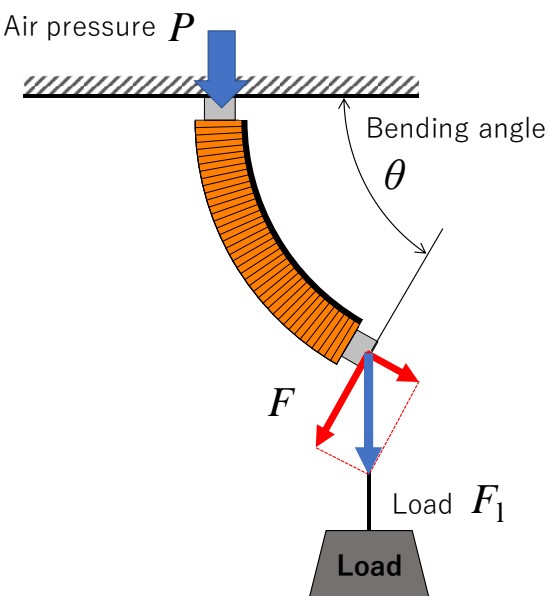

**Figure 6.** Schematic of the static characteristics experiment for BPAM.

The experimental results and theoretical lines are shown in Figure 7. Furthermore, Table 2 shows the mean square error (MSE) between the experimental results and the theoretical line at each pressure. From Figure 7, though the static characteristics model shared a similar tendency with the experimental results, errors increased in the high- and low-pressure domain. This is also confirmed in Table 2. We considered that this was because of the dead-band characteristic of BPAM at low pressure and irregular

bending at high pressure. Although the BPAM was regarded as bending in plane, as shown in Figure 6, the actual BPAM moved along the depth direction as it was not constrained by any rigid structure when the BPAM bent through a large angle due to high pressure. Moreover, the BPAM developed manually by hand may twist because of the crooked arrangement of the aramid fibers, which were aligned in the longitudinal direction. This theoretical model will be used for simulation and force control of the SDR in future works. To improve the usability of the model, it was necessary to include dead-band characteristics and the operable range into the model.

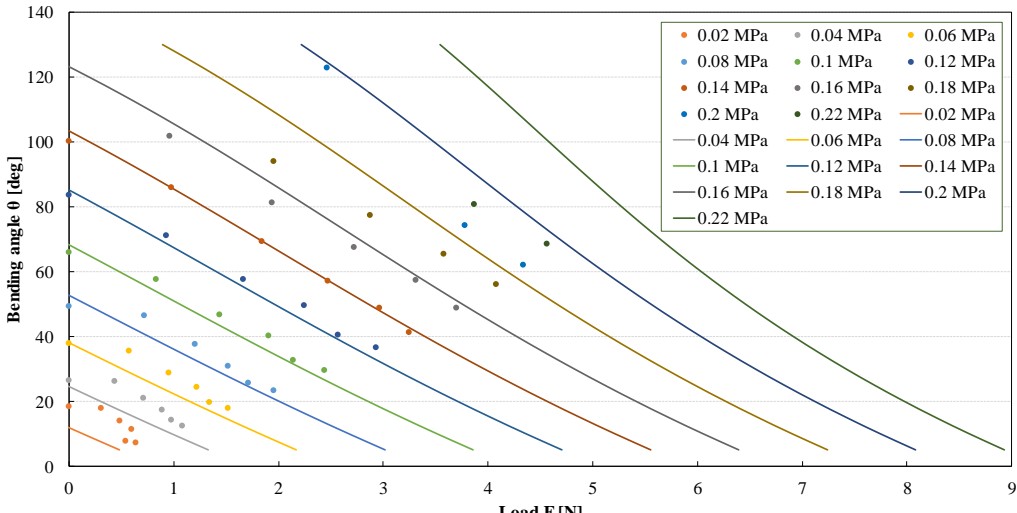

**Figure 7.** Results of the static characteristics experiment and theoretical lines from a BPAM model.

**Table 2.** MSE of the static characteristics experiment results and theoretical lines.

| Air Pressure (Mpa) | MSE | Air Pressure (MPa) | MSE |
|---|---|---|---|
| 0.02 | 56.87 | 0.14 | 2.17 |
| 0.04 | 34.23 | 0.16 | 15.39 |
| 0.06 | 22.33 | 0.18 | 115.61 |
| 0.08 | 14.77 | 0.20 | 202.07 |
| 0.10 | 11.77 | 0.22 | 1316.83 |
| 0.12 | 9.13 | | |

## 3. Self-Deformation Robot (SDR)

### 3.1. Robot Design

The SDR with BPAMs as flexible frames is displayed in Figure 8. Its structure was identical to a wire frame and assumed a cubical shape as the SDR developed for the initial test. The cubic SDR had a side 200-mm long and weighed 0.7 kg. The BPAMs were arranged as edges and were all connected to each corner. Designed with an omnidirectional structure, aramid fibers limiting the axial length of the BPAMs were facing the cube center, which allowed for SDR expansion toward the outer side. Unactuated BPAMs were bent in various directions relative to the actuated BPAMs. Consequentially, the SDR could deform itself by bending, twisting, or getting flattened, a behavior unique to soft-actuator-driven soft robots.

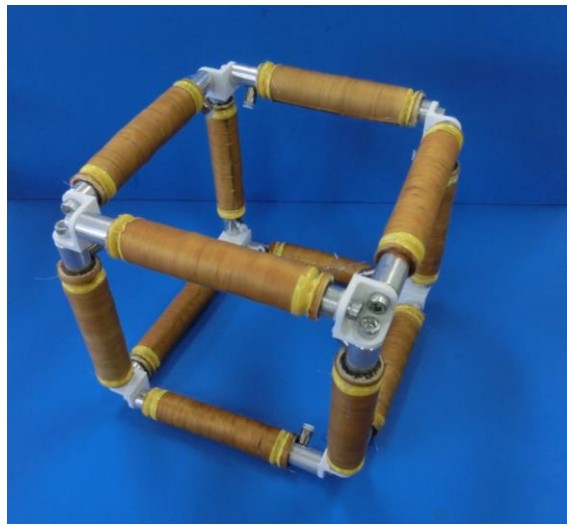

**Figure 8.** The self-deformation robot.

Moreover, the control system of the SDR is shown in Figure 9. Each BPAM of the SDR was applied with compressed air from a compressor through a three-port solenoid valve controlled by an Arduino mega through a MOSFET switching circuit. In turn, the Arduino mega received command signals (on/off for each BPAM) from a personal computer via a serial connection.

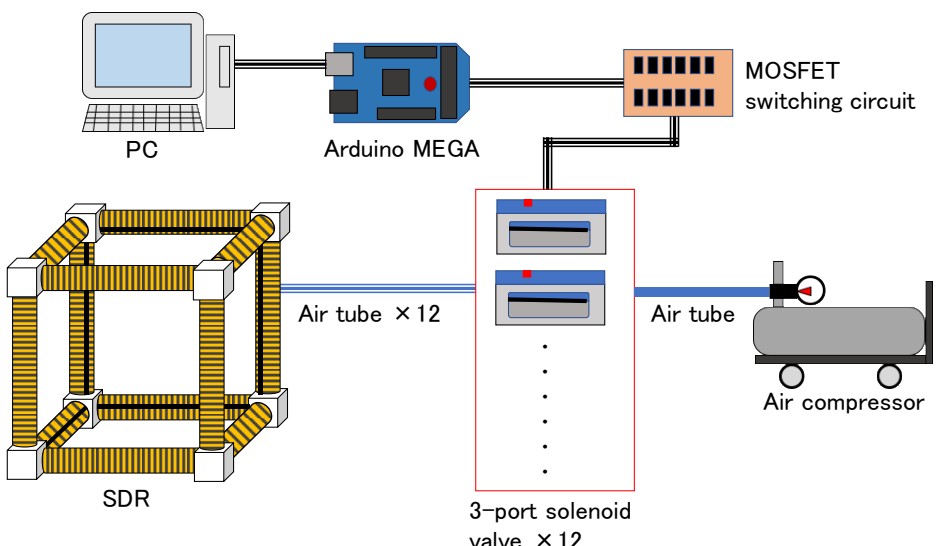

**Figure 9.** Control system of the self-deformation robot (SDR).

*3.2. Proposed Locomotion Method*

We first needed to identify the BPAMs to explain the locomotion of the SDR. Each corner used for identification is shown in the upper side of Figure 10. For instance, the BPAM connected by end corners, A and B, could be defined as BPAM(AB). Owing to the complex nature of the SDR deformation, conventional rigid-body dynamics may be unsuitable for the robot. We are planning to address this problem in the future using a simulation. Nevertheless, for this paper, we proposed a locomotion method established by trial-and-error as the first step. When the SDR moved in the direction of the arrow in Figure 10, the BPAMs were controlled, as shown in the lower side of Figure 10. Here, the air pressure applied to the BPAMs was set at 0.3 MPa:

- BPAM(BF) and BPAM(CG) bent (1).

- BPAM(AB), BPAM(CD), BPAM(EF), and BPAM(GH) bent (2).
- Air pressure in BPAM(BF) and BPAM(CG) was released; these BPAMs become unactuated (3). Next, the SDR rolled (4).
- Finally, air pressure in BPAM(AB), BPAM(CD), BPAM(EF), and BPAM(GH) was released.

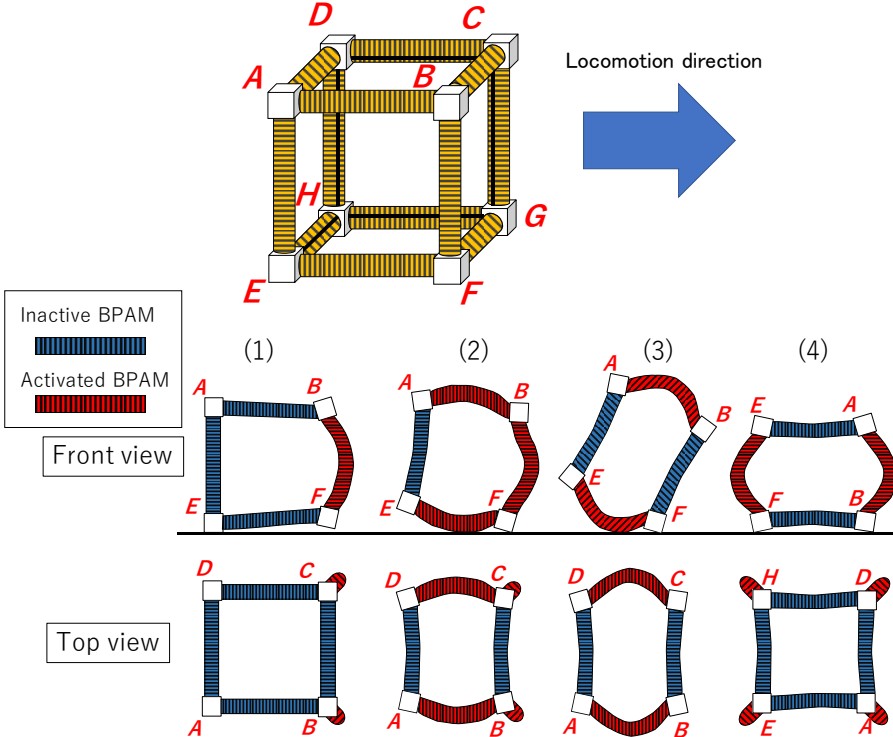

**Figure 10.** Identification of BPAMs of the SDR, along with the locomotion sequence.

Numbers from (1) to (4) in the above explanations correspondent to the sequence numbers in Figure 10.

Using this procedure, the SDR rolled in such a way that the plane face of BCGF shifted to the bottom. Step 1 was considered when the SDR's center of gravity tilted toward the locomotion direction, followed by Step 2, where it became unstable and rolled over to the direction along which the center of gravity tilted in Step 1. As the SDR was an omnidirectional structure, it could continue moving on the same manner.

## 4. Locomotion Demonstration

Subsequently, we undertook a locomotion test for the SDR. The SDR locomotion cycle using a locomotion approach explained in Section 3.2 of Chapter 3 is shown in Figure 11. Here, phases 1 to 2 correspond to Step 1 of the locomotion method, phases 2 to 3 for Step 2, phases 3 to 5 for Step 3, and phases 5 to 6 for Step 4. It could be observed that the SDR achieved locomotion by deforming itself. In particular, BPAM(AE) and BPAM(DH) were bent sharply by other activated BPAMs in phases 4 and 5. Although allowed based on the structure, such deformation left a little damage on the BPAMs. As such, this sharp bending should be avoided as much as possible in view of the lifetime of the BPAMs.

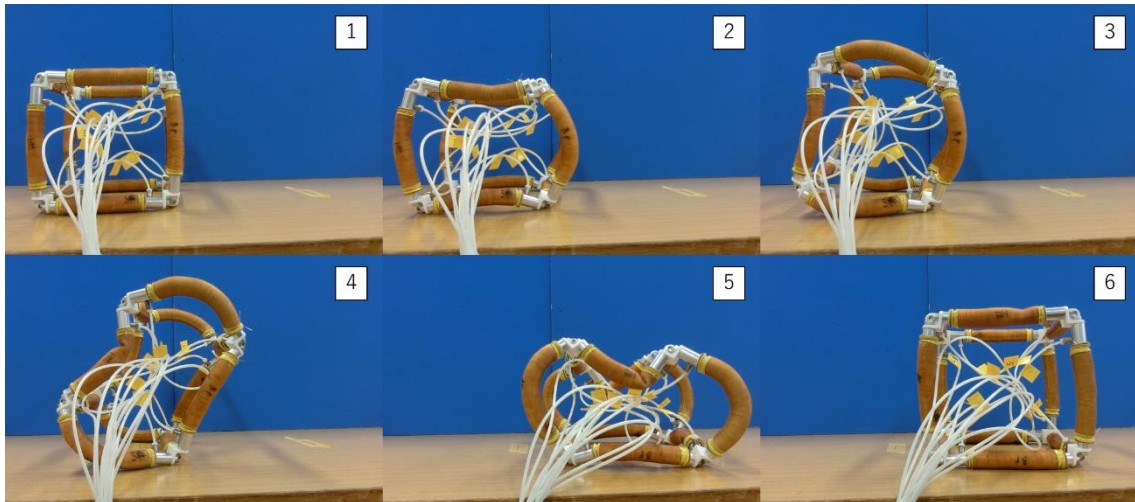

**Figure 11.** Locomotion of the SDR.

Accordingly, we conducted a simplified experiment to measure the locomotion speed of the SDR. Here, we measured the time for each step and movement distance, as shown, respectively, in Figure 12a,b for one locomotion cycle. Afterward, we calculated the locomotion speed from the time and distance data. We conducted locomotion experiments 10 times. Then, the average time, distance, and speed from them were calculated.

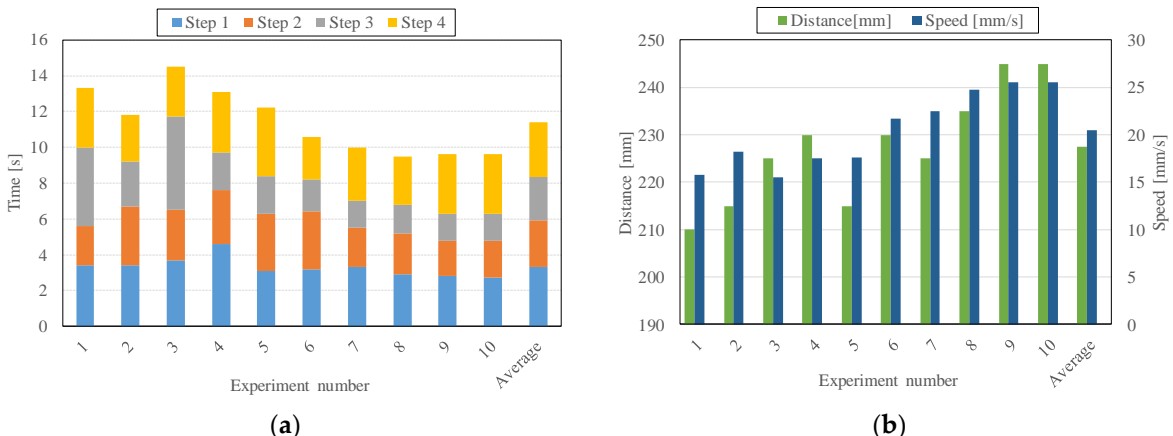

**Figure 12.** Measurement results of locomotion speed. (**a**) Time for each step; (**b**) Movement distance and locomotion speed.

The movement distances were almost identical to the side length. Owing to the slips and bounds in Step 3, the distances were often longer than 200 mm, and the step time varied. Moreover, although there was variability in locomotion speed, the average speed was 20 mm/s, which is reasonable for our application.

## 5. Discussion

By conducting the locomotion demonstration seen in Chapter 4, we observed some interactions between BPAMs in the SDR. While an activated BPAM bent an adjacent inactive BPAM (phases 2 and 4 in Figure 11), two activated BPAMs suppressed the bending of each other (phase 3 in Figure 11). Each BPAM generated a bending moment that bent adjacent BPAMs inward when activated. Since a BPAM activated with a high pressure resulted in a sharp flexure of other BPAMs, we need to control the intensity of interactions. For example, the sharp flexure of an inactive BPAM can be suppressed by applying air pressure to that BPAM. However, this method does not change the intensity of

interaction, and the whole stiffness of the SDR will increase due to activated BPAMs bending each other antagonistically. On the other hand, a decreasing stiffness of corner parts can decrease the intensity of interactions. Flexible corner parts absorb the bending moment from activated BPAMs and will prevent sharp flexure of adjacent inactive BPAMs. Although this method reduces the load of the SDR, interactions that have too low a pressure may result in a reduction of deformation of the SDR. Appropriate stiffness of each part of the SDR, and not just the corner parts, need to be decided upon for improving performance. Due to the difficulty of applying rigid-body dynamics to the SDR, designing stiffness of the body using simulations is regarded as useful. An improved model of the BPAM will help us to develop the simulation.

## 6. Conclusions

For this paper, we developed an SDR using BPAMs. We provided a description and discussion for the BPAMs and conducted a static characteristics experiment, in which the BPAM model exhibited a similar tendency with the experimental results, and the actual BPAM particularly moved along the direction of depth. Moreover, we proposed and demonstrated an effective method of locomotion for the SDR. We calculated the locomotion speed by measuring the drive time and movement distance. For the primary objective of monitoring children and the elderly, the SDR's speed was considered reasonable. However, during the demonstration, some BPAMs were bent sharply by other activated BPAMs during the SDR driving process, leaving a little deformation on these BPAMs.

In the future, we are planning to improve the BPAM structure to prevent sharp bending and unnecessary twists while the SDR is driving, along with the possibility to equip the SDR with built-in control systems, such as a compressor. Additionally, we are planning to mount sensors, including a tactile sensor and an attitude sensor, and a camera on the SDR for practical use. We are also planning to optimize the SDR motion and body design via simulation. Then, we are also planning to use simulations for locomotion planning. Improving the theoretical model, such as including operable range into the model, is necessary to achieve it.

**Author Contributions:** Conceptualization, H.T. and K.H.; software, H.T.; validation, H.T. and N.I.; formal analysis, S.K.; investigation, N.I. and T.I.; writing—original draft preparation, H.T.; visualization, H.T.; supervision, H.T.; project administration, H.T.

**Funding:** This research was supported by the Program on Open Innovation Platform with Enterprises, Research Institute and Academia (OPERA) from Japan Science and Technology Agency (JST) (grant no. JPMJOP1614).

**Acknowledgments:** The bending-type pneumatic artificial muscles used aramid fiber provided by Teijin Ltd., to which we would like to express gratitude.

**Conflicts of Interest:** The authors declare no conflict of interest.

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
