# Peer review of "A Self-Deformation Robot Design Incorporating Bending-Type Pneumatic Artificial Muscles"

_technologies, doi:10.3390/technologies7030051_

Round 1
Reviewer 1 Report
The paper presents mathematical modelling, robotic hardware design and testing of a soft robot actuated by pneumatic artificial muscles. The work is submitted as article but the manuscript is short and the description synthetic.
The locomotion of self-deformable robots is quite important in the field of autonomous robotics and therefore this research should be relevant. However, the manuscript weakly explain the main parts of the contribution to the field.
I have the following revision/comments to be considered:
The derivation of (1)-(3) is not straightforward. The addition of the step-by-step procedure should be very helpful;
I would strongly recommend authors checking the goodness-of-fit of the model used for identification of the force data of Fig.4. The fit of the model is poor at low and high force values. It is not clear how this model is used for locomotion control;
Fig. 9: an average is displayed. What data averaged? Why, in a single experiment?
Author Response
We appreciate your review and comments. We have revised our paper according to suggestion. Please check modifications below.
>The derivation of (1)-(3) is not straightforward. The addition of the step-by-step procedure
>should be very helpful;
Thank you for suggestion. I added procedure of derivation in section 2.2.
>I would strongly recommend authors checking the goodness-of-fit of the model used for
>identification of the force data of Fig.4. The fit of the model is poor at low and high force
>values. It is not clear how this model is used for locomotion control;
Thank you for recommendation.
We calculated mean square error in each line to check the goodness-of-fit. Errors increased at high- and low- pressure domain. We considered that is because of dead-band characteristic of BPAM at low pressure and irregular bending at high pressure.
We conducted the static characteristics experiment to understand BPAM’s basic characteristics. We are planning to improve the model. Then model will be used for simulation and force control as future works. We added this explain to chapter 2 and conclusion as future works.
>Fig. 9: an average is displayed. What data averaged? Why, in a single experiment?
Thank you for question. We conducted locomotion experiments 10 times. Then average time, distance and speed from them were calculated. We added this explanation in chapter 4.
Reviewer 2 Report
I read through the paper, I would be happy to reassess the paper for its suitability for publication should the authors address the following comments:
- The introduction is very narrow. Why PAM actuators are used? What do PAMs offer that other actuators don't? Recent review papers on artificial muscles can give the authors some clue.
- Page 3 - The authors should show the steps they took to derive the equations. If they are using those from somewhere else, they should cite the source appropriately.
- I recommend the authors make a table and compare the characteristics of their bending actuator with other bending actuators (e.g., IPMCs, Conducting Polymers, Nylon Bending actuators, other pneumatic actuators, etc).
- The reference list is very narrow. More papers (from different countries) relevant to this field should be cited.
Author Response
We appreciate your review and comments. We have revised our paper (added references, introduction etc.). Please check modifications below.
>- The introduction is very narrow. Why PAM actuators are used? What do PAMs offer that
>other actuators don’t? Recent review papers on artificial muscles can give the authors some
>clue.
Thank you for suggestion. We added reasons for using PAM with some references in introduction.
>- Page 3 - The authors should show the steps they took to derive the equations. If they are
>using those from somewhere else, they should cite the source appropriately.
Thank you for suggestion. I added procedure of derivation in section 2.2.
>- I recommend the authors make a table and compare the characteristics of their bending
>actuator with other bending actuators (e.g., IPMCs, Conducting Polymers, Nylon Bending
>actuators, other pneumatic actuators, etc).
Thank you for recommendation. I tried to make the table to compare these actuators. However, it was difficult to make unified comparison because there were actuator’s data under various condition, scale, experimental method.
So, I added general comparison between these actuators and pneumatic actuator in introduction using reference that compared these actuators. And I also added why pneumatic actuator were used for SDR based on the comparison.
>- The reference list is very narrow. More papers (from different countries) relevant to this
>field should be cited.
Thank you for advice. I added references to explain advantages of soft robot in introduction.
Reviewer 3 Report
Please check attached file.

Author Response
We appreciate your review and comments. We have revised our paper (explanation of the BPAM, sequence of the SDR with illustrations, etc. ). Please check modifications below.
>1. There is not a chapter of “Discussion” which is very important part in journal paper. You >should discuss about the results in Discussion section.
Thank you for advice. I made new chapter for discussion and discuss about locomotion demonstration of the SDR.
>2. In this paper, there are few references as for a journal paper. I suggest to add references.
Thank you for suggestion. Due to revision of introduction, references were added.
>3. At line 193 in Section “References”, Miyagawa’s paper was not written in English. If the
>literature were not written in English, this is not very useful for many readers of an English
>journal and I think you should provide the reader with a short summary of the cited study
>for allowing many readers to check the reference. Moreover, I think that the description “(in
>Japanese)” is necessary in the reference [10].
Thank you for advice. I added short summary of this reference in introduction. Also I added description “(in Japanese)” to the reference.
>4. At line 73-76, although you explained about BPAM, I think that explanations and
>illustrations are insufficient. I suggest to add explanations and illustrations about the motion
>principle and mechanism of BPAM such as a pitch and thickness of aramid fibers in
>circumferential and longitudinal direction for comprehensibility.
Thank you for suggestion. I added illustration and text to explain drive principle of BPAM in section 2.1.
>5. You illustrated an equilibrium between applied air pressures P, load F, and bend angle θ
>by equation (1) and (2). However, it is not easy to understand the equilibrium by using only
>the equation (1), (2) and Figure 2. Moreover, I can’t find the explanation of “Mgr”. On the
>other hand, it is easy to understand similar model in Miyagawa’s paper in Reference [10]
>for detailed description (Equation (1)-(9) and Figure 1-4). I think that current explanation
>of the model with only equation (1), (2) and Figure 2 is insufficient, and many reader around
>the world can’t read the Miyagawa’s paper as described above. Therefore sufficient
>explanation with illustration for the model of BPAM is necessary. I suggest to add that.
Thank you for suggestion. I added procedure of derivation and illustrations of BPAM in section 2.2.
>6. In addition, at line 124-128, you mentioned a sequence of BPAM’s actuation. But it is not
>easy to understand the mechanism of robot motion. Therefore, you should also explain the
>mechanism and sequence with illustrations like the Figure 8and add that in Figure 7.
>Moreover, I suggest to illustrate that at front view and top view(from two directions) for
>comprehensibility of each BPAM’s bending direction because each BPAM seem to bend
>three-dimensionally.
Thank you for suggestions. I added illustrations of locomotion sequence of SDR in the Figure you pointed. Additional illustrations consist of front view and top view.
Round 2
Reviewer 1 Report
The paper can be accepted in present form.
Reviewer 2 Report
Good work.
Reviewer 3 Report
I think that this version is significantly improved and can be accepted in present form.